# Signal Deconvolution and Generative Topographic Mapping Regression for Solid-State NMR of Multi-Component Materials

**DOI:** 10.3390/ijms22031086

**Published:** 2021-01-22

**Authors:** Shunji Yamada, Eisuke Chikayama, Jun Kikuchi

**Affiliations:** 1Graduate School of Bioagricultural Sciences, Nagoya University, Furo-cho, Chikusa-ku, Nagoya 464-8601, Japan; shunji.yamada@riken.jp; 2Environmental Metabolic Analysis Research Team, RIKEN Center for Sustainable Resource Science, 1-7-22 Suehiro-cho, Tsurumi-ku, Yokohama 230-0045, Japan; chikaya@nuis.ac.jp; 3Department of Information Systems, Niigata University of International and Information Studies, 3-1-1 Mizukino, Nishi-ku, Niigata 950-2292, Japan; 4Graduate School of Medical Life Science, Yokohama City University, 1-7-29 Suehiro-cho, Tsurumi-ku, Yokohama 230-0045, Japan

**Keywords:** solid-state NMR, short-time Fourier transform, signal deconvolution, prediction, anisotropy, *T*_2_ relaxation, macromolecules, cellulose degradation, plastics, *Euglena gracilis*

## Abstract

Solid-state nuclear magnetic resonance (ssNMR) spectroscopy provides information on native structures and the dynamics for predicting and designing the physical properties of multi-component solid materials. However, such an analysis is difficult because of the broad and overlapping spectra of these materials. Therefore, signal deconvolution and prediction are great challenges for their ssNMR analysis. We examined signal deconvolution methods using a short-time Fourier transform (STFT) and a non-negative tensor/matrix factorization (NTF, NMF), and methods for predicting NMR signals and physical properties using generative topographic mapping regression (GTMR). We demonstrated the applications for macromolecular samples involved in cellulose degradation, plastics, and microalgae such as *Euglena gracilis*. During cellulose degradation, ^13^C cross-polarization (CP)–magic angle spinning spectra were separated into signals of cellulose, proteins, and lipids by STFT and NTF. GTMR accurately predicted cellulose degradation for catabolic products such as acetate and CO_2_. Using these methods, the ^1^H anisotropic spectrum of poly-ε-caprolactone was separated into the signals of crystalline and amorphous solids. Forward prediction and inverse prediction of GTMR were used to compute STFT-processed NMR signals from the physical properties of polylactic acid. These signal deconvolution and prediction methods for ssNMR spectra of macromolecules can resolve the problem of overlapping spectra and support macromolecular characterization and material design.

## 1. Introduction

Recently, research for a low-carbon society has gained importance from the viewpoints of global challenges such as the marine pollution of marine plastics, waste disposal, and global warming [1]. Microbial products and plant biomass as alternatives to petroleum resources can be used to produce macromolecular materials such as plastics and feedstock [2]. Polymers such as polylactic acid (PLA) [3], poly-ε-caprolactone (PCL) [4], and cellulose [5,6,7,8,9,10,11,12] are multiple domain/component systems and are often employed as high-performance materials with various properties. Microbial and plant biomass should be analyzed as a biochemical system composed of multiple components containing macromolecules with multiple domains. Solid-state nuclear magnetic resonance (ssNMR) spectroscopy is a powerful tool for characterizing the native structure, components, and dynamics of solid-state samples at the atomic level. It is being increasingly applied in material/life sciences [13,14]. Therefore, an advanced ssNMR analytical approach must be developed for macromolecular products such as microbial products, plant biomass, and plastics.

Various techniques that use high-field magnets, cryogenic detection systems, indirect detection [15], nonuniform sampling [16], and dynamic nuclear polarization methods [17,18] have been developed for realizing increased sensitivity. From the aspect of NMR measurement, various solid-state NMR methods have been used. Typical methods are cross-polarization (CP)–magic-angle spinning (MAS) methods, static multiple-quantum (MQ) NMR, static ^1^H NMR [19], direct polarization (DP), high-resolution (HR)-MAS [20,21,22], magic-and-polarization echo (MAPE) filtering [23], double-quantum (DQ) filtering [24], and combined rotation and multiple-pulse techniques (CRAMPS) [25]. MAS probes are capable of spinning frequencies much greater than 100 kHz [26]. Other advanced techniques are spin diffusion measurements [27], pulsed field gradient (PFG) NMR, diffusion-ordered spectroscopy (DOSY), and time-domain NMR/relaxometry [28]. In addition, multi-dimensional NMR was applied for separating overlapping spectra; examples of such techniques are wide-line separation (WISE) and heteronuclear correlation (HETCOR) [29,30], three-dimensional (3D) dipolar-assisted rotational resonance, double-cross-polarization ^1^H-^13^C correlation spectroscopy, and ^1^H–^13^C solid-state heteronuclear single-quantum correlation spectroscopy [22].

In the characterization of solid-state samples with crystal, interphase, and amorphous domains, the anisotropy detected by static measurement is useful, but its analysis is difficult because the spectra are broad and overlapping [31]. Therefore, the application of signal deconvolution to measure solid-state NMR data is an important challenge to extract hidden information in the NMR spectra of macromolecular samples with multiple phases and components. Several methods for spectral separation [32], apodization, zero filling, linear prediction, fitting and numerical simulation [33], such as covariance analysis [34], SIMPSON [35], SPINEVOLUTION [36], dmfit [37], EASY-GOING deconvolution [38], INFOS [39], Fityk [40], ssNake [41], the noise reduction method based on principal component analysis [42], and the signal deconvolution method that combines short-time Fourier transform (STFT, a time–frequency analytical method), and probabilistic sparse matrix factorization (PSMF which is one of the non-negative matrix factorizations) [43] were developed as computational approaches to measured data.

In this study, we propose signal deconvolution methods using STFT and non-negative tensor/matrix factorization (NTF, NMF) optimized to characterizing the solid-state NMR spectra of macromolecular samples with multiple domains and components such as cellulose, plastics, and *Euglena gracilis*. Using generative topographic mapping regression (GTMR, the regression method using GTM) [44], we mutually predicted higher-order structure descriptors of STFT-processed NMR signals (STFT–NMR signals) and physical properties of the material. To the best of our knowledge, this is the first reported application on the prediction of NMR signals from the thermal properties of plastics using GTMR.

## 2. Results and Discussion

### 2.1. Signal Deconvolution and Prediction for Solid-State NMR of Multi-Component Materials

In this study, from a practical point of view, we focused on a signal deconvolution method for one-dimensional (1D) ssNMR data suitable for high-throughput multi-sample measurement. In particular, static ^1^H anisotropic spectra can be used as an index of the motility of higher-order structures, but these spectra are broad and show overlapping. Even extremely sharp spectra such as ^13^C CP-MAS show overlaps, especially in the case of signals with different mobility derived from the same atom. Therefore, those data must be separate signals. In principle, the exponential decay constant of the free induction decay (FID) obtained by applying a 90° pulse to create transverse magnetization is the *T*_2_ relaxation time. In reality, however, because of the effect of magnetic field inhomogeneity, the decay constant of the FID is defined as *T*_2_^*^, an instrument-dependent parameter, rather than *T*_2_. In this paper, we report a signal deconvolution method to separate the broadening spectra derived from macromolecules (cellulose and plastics) with multiple phases and components based on the *T*_2_^*^ relaxation pattern. The short-time Fourier transform (STFT) method is used to convert an FID into frequency domain data at short time intervals to generate a matrix of time and frequency axes (Figure 1a). As algorithms of factorization, in addition to the traditional NMF for analysis of the two-dimensional (2D) dataset, we investigated the application of NTF (non-negative Tucker decomposition (NTD) [45] and non-negative canonical polyadic decomposition (NCPD) [46,47]), which is a factorization algorithm useful for the analysis of the 3D dataset of multiple samples and parameters. By applying NTF/NMF (Appendix A) to the dataset, the signal components were separated based on the *T*_2_^*^ relaxation pattern of the components indicated in the multi-phase and multi-component spectra (Figure 1b,c). Furthermore, the high-order structure of materials exerts a significant influence on their macroscopic properties [27]. Traditional design approaches for materials are experimentally driven and trial-and-error are facing significant challenges due to the vast design space of materials. In addition, computational technologies such as density functional theory (DFT) [48] and molecular dynamics (MD) [7] are usually computationally expensive and are difficult to calculate molecular structures from material properties. To address these problems, machine-learning-assisted materials design is emerging as a promising tool for successful breakthroughs in many areas of science [49]. In addition, NMR measurement, especially a low magnetic field NMR, is a method for routine material evaluations, which produce a lot of NMR datasets [32]. Against this background, in the cycle of developing materials using NMR and other measurements, the prediction of the NMR signal using the accumulated data is necessary to find a structure with the desired properties. In this study, prediction of the NMR data and sample properties was calculated using GTMR (Figure 1d,e and Appendix A) [44]. For cellulose degradation samples, our previous study reported that solution ^1^H and ^13^C NMR data were used for evaluating the concentration of catabolic products. In this study, we examined the use of pseudodata as a method of predicting data without experiments. Pseudodata are a dataset with the same distribution as the original dataset generated using Gaussian mixture models (GMM) (Appendix A) [50]. Randomly generating data based on means and covariances using GMM produces new pseudodata. By performing GTMR calculation from these pseudodata as input data, a spectrum as output can be predicted without preparing new materials. The STFT–NMR signals were predicted as a higher-order structure descriptor and were transformed to predicted NMR properties. This method can be applied to various sample systems for pursuing structure–property correlation. In this study, we demonstrate the application of cellulose degradation and plastic for evaluating our method. Here, in cellulose degradation, the word “higher-order structure” means the crystalline and amorphous structure of cellulose, and the word “property” means the quantity of catabolic products. In addition, with plastics such as PCL, it is difficult to design those having both high degradability and toughness. In the PCL, multiple domain structures with different degrees of entanglement of molecular chains are referred to as “higher-order structures”, and thermal and mechanical properties are referred to as “property”. This analytical flow is useful for the research and development of macromolecules and related products.

### 2.2. Non-Negative Tucker Decomposition to ^13^C CP-MAS in Cellulose Degradation Process

Solid and solution NMR methods can monitor higher-order structural changes and catabolic products during the degradation of cellulose by microorganisms [10,12]. The dataset used in Figure 2 is a time-dependent dataset of ^13^C solid-state CP-MAS signals of the cellulose degradation process and also contains signals of catabolic products (proteins and lipids). The ^13^C ssNMR spectra detect macromolecules of cellulose, proteins, and lipids. This dataset is a set of data with frequency and intensity in 16 time points from 0 to 120 h (Figure 2a). This dataset was processed by STFT (Appendix A). We demonstrated the application of NTD (Figure 1b or Figure 2b), which is one of the tensor factorizations for multi-sample data. By separating the spectrum into four components, it was possible to visualize the spectral patterns (Figure 2c–f), time change of each component (Figure 2g), and the composition (Figure 2h). The word “Time change” in Figure 2g means the change in acquisition time of the separated signal components. In addition, The word “Composition” in Figure 2h means the change in the 16 samples from 0 to 120 h of ^13^C CP-MAS NMR spectra. As a result, the four signals (the cellulose, proteins, and lipids-like signals) were clearly separated as intense signals, while the noise was relatively low. In the calculation scheme of NTD, the convergence tolerance of calculation error was less than 0.001. The cellulose-like spectrum had a short relaxation time (Figure 2c,g (orange)), the protein-like spectrum had a long relaxation time (Figure 2d,g (green)), and the lipid-like spectrum had the longest relaxation time (Figure 2e,g (red)); the noise did not change. It was possible to evaluate the concentration of each component among samples (Figure 2h). As a result of separating the spectrum of the cellulose C4 region (Appendix A) into six components, it was possible to visualize the spectral patterns (Appendix A), time change of each component (Appendix A), and the composition in each sample (Appendix A). So far, tensor factorizations have been reported for the application of NCPD to solution NMR of carbohydrate mixtures [46] and high-dimensional NMR of protein structures [47]. As a result of separating the spectrum into four components using NCPD, it was not as good as NTD because of unclear spectral patterns for assigning compounds (Appendix A). NCPD is different from the algorithm of NTD used in this work. NTD separates the tensor into a small core tensor and factor matrices. NCPD separates the tensor into factor matrices without a core tensor. This study shows that the NTD is also effective for analyzing time-series ssNMR data such as those of the cellulose degradation process.

### 2.3. Non-Negative Matrix Factorization to Static ^1^H ssNMR in PCL and E. gracilis Samples

PCL has a high-order structure of mobile, rigid, and interphase [28,33]. Evaluating the structure, motility, and proportion of multiple domains is important for material development including such as the optimization of physical properties. In the development of plastics especially, evaluation of higher-order structures is useful for the static ^1^H anisotropic spectrum in solid states. From the aspect of the pulse program, by using a DQ filter or MAPE filter, components with different motilities can be extracted. In this study, we demonstrated the application of NMF to a 2D dataset created from the single data of PCL using STFT. Unlike NTF for a 3D dataset mentioned above, NMF is a method for a 2D dataset. NMF discovers hidden patterns in the axes of both time and frequency created by STFT, which is able to separate NMR signals to multiple components with different *T*_2_^*^. It was shown that by using NMF, rigid and mobile phases can be extracted from a broad static ^1^H anisotropic spectrum of PCL as the components related to different physical properties (Figure 1c and Figure 3). We resolved the linear macromolecular structure as a mobile domain and the branched macromolecular structure due to strong anisotropic ^1^H-^1^H dipolar coupling as a rigid domain in solid material such as PCL. Furthermore, we demonstrated this method for ^1^H, ^13^C, ^15^N and ^31^P spectra of microalgae such as *E. gracilis* in a multi-component system (Appendix A). ^1^H high-speed magic-angle spinning (MAS) spectrum was separated into signals of amide protons and fatty acids in lipids, and the ^13^C CP-MAS spectrum was separated into signals of paramylon, lipids, and proteins. To overcome the limitation of sensitivity in NMR, various techniques were developed using high-field magnets, cryogenic detection systems, indirect detection [15], nonuniform sampling [16], and dynamic nuclear polarization methods [17]. We previously demonstrated that the STFT can be used for signal improvement of the solution diffusion-edited NMR spectra, including broad signals and sharp signals [43]; in this study, we demonstrated signal deconvolution using the STFT in the solid-state NMR. When using this method for NMR data with low digital resolution such as solid-state NMR and quadrupole nucleus, this signal deconvolution method needs additional efforts. We demonstrated some interpolation methods for increasing data points (Appendix A). The Fourier interpolation method provides an interpolated spectrum without artifact signals. Spectra interpolated by other methods have artifacts in the extended region.

### 2.4. Prediction of Concentration of Products in the Cellulose Degradation Process

Thus far, GTM has been applied to characterize NMR data [51]. Recently, computational approaches for predicting NMR signals [48], chemical structures [52], and physical properties [53,54,55,56,57] were developed. Chemical shifts of NMR are rich in chemical information and enable encoding the structural features of the molecules contributing to their physical/chemical/biological properties. Thus, it has potential for use as a descriptor in quantitative structure–activity/property relationship (QSAR/QSPR) modeling studies [58]. GTMR was applied for analyzing these studies [44]. Therefore, the prediction of NMR signals is important for developing materials. This study is the first application of GTMR for the prediction of NMR signals (Figure 1d). In the degradation of cellulose, cellulose is metabolized into microbial cell components such as proteins and lipids, and then catabolized into short-chain fatty acids. In Figure 2, macromolecules (cellulose, proteins, and lipids) were detected using the solid ^13^C spectrum. In addition, to track the process of material degradation, solution NMR spectra were used to detect small molecules such as propionate and acetate. Therefore, the catabolic products were captured by solution NMR (the final product is CO_2_ and CH_4_ with one carbon atom (Appendix A)). During GTMR, multi-dimensional and multi-component data (in this case, CP-MAS macromolecular data and small-molecule solution NMR data) can be mapped into the reduced dimensional space (Figure 4a,b left). When cellulose is finally catabolized to CO_2_ by the catabolism of microorganisms, it is metabolized into acetate with two carbon atoms and CO_2_ with one carbon atom via propionate with three carbon atoms. When the signal intensity of propionate is used as the input data of GTMR, it is possible to predict both the properties (scaled signal intensities in these results) of acetate (Figure 4a right; R^2^ = 0.976) with the two carbon in the previous stage of the final product and CO_2_ (Figure 4b right; R^2^ = 0.967) with one carbon in the final product. GTMR thus provides information about the predicted NMR scaled signals of products in cellulose degradation. This information is important for monitoring the degradation process due to a key in compound production using cellulose.

### 2.5. Prediction of NMR Signals from Thermal Properties in Plastics

This study is the first application to predict NMR signals from the thermal properties of plastics using GTMR. The design method for higher-order structures of plastics should control the glass transition, melting, and degradation temperature (*T*_g_, *T*_m_, and *T*_d_) as thermal properties. The GTMR was first applied for the inverse analysis of the CP-MAS spectra (Appendix A) from the thermal properties (Appendix A) of PLA in the solid state (Figure 1e). Therefore, *T*_g_ (Figure 5a), *T*_m_ (Figure 5b), and *T*_d_ (Figure 5c) were mapped into a reduced 2D space. We focused on the prediction of the intended thermal property (Figure 5d; red cross) using the three GTMR maps (*T*_g_, *T*_m_, and *T*_d_). Hence, the STFT–NMR signals, i.e., the predicted spectrum, corresponded to the red cross and were predicted as higher-order structure descriptors (Figure 5e). Moreover, as a result of predicting the thermal properties from pseudo-CP-MAS spectra of PCL using GMM, it was possible to predict thermal properties (Appendix A).

Recently, the materials informatics (MI) approach was considered for material design [59] because the intended physicochemical property is really hard to identify in the material development process. Therefore, the MI approach uses “big-data” such as deposited database, as well as monitoring and analyzing higher-order structural data during the materials production process [60,61]. When developing a material with the desired physical properties, the molding conditions of the material with the predicted structure play an important role.

## 3. Materials and Methods

### 3.1. NMR Analysis

The ssNMR data were acquired using an Avance III HD-500 spectrometer (Bruker Corp., Billerica, MA, USA) equipped with a double-resonance 4.0 mm MAS probe. The solution NMR data were acquired using an Avance III HD-700 spectrometer (Bruker Corp., Billerica, MA, USA). The ^1^H and ^13^C CP-MAS spectra and solution ^1^H and ^13^C NMR spectra of cellulose previously reported by Yamazawa et al. were used [10]. The multiple phases polymer such as PCL, were measured using static, MAPE-filtered and DQ-filtered ssNMR. The ^1^H, ^13^C, ^15^N, and ^31^P spectra of *E. gracilis* cell previously reported by Komatsu et al. were used [22].

### 3.2. Thermal Analysis of Plastics

Thermogravimetry (TG) and differential thermal analysis (DTA) measurements were conducted using an EXSTAR TG/DTA 6300 (SII NanoTechnology Inc., Tokyo, Japan) instrument [29,62]. Approximately 10 mg of samples was individually vaporized at 5 °C/min from 40 to 500 °C in a nitrogen atmosphere. The *T*_m_ and *T*_d_ were determined as the endothermic peak in DTA curves and the peak of weight loss in Derivative Thermogravimetry (DTG) curves. Differential scanning calorimetry (DSC) was conducted using a DSC3500A (NETZSCH Geratebau GmbH, Selb, Germany) [63]. Approximately 1.5 mg of samples was individually measured at the following steps at 10 °C/min from 25 to −30 °C, at 10 °C/min from −30 to 200 °C, and at 20 °C/min from 200 to 25 °C in a nitrogen atmosphere. The *T*_g_ was determined as an endothermic peak during heating.

### 3.3. Signal Deconvolution Methods

The signal deconvolution method was developed in Python 3. The processing of NMR data was implemented by using the nmrglue [53] package in Python. Tensor factorization methods of NTD and NCPD were calculated using TensorLy Python library for tensor methods [45], and NMF was calculated based on the NIMFA Python library for non-negative matrix factorization [64]. NMR data with interpolated data points were created using “signal” and “interpolate” in “scipy”.

### 3.4. Prediction Methods

Predictions of NMR signals and properties were calculated using GTMR [44]. In the analysis of cellulose degradation, a regression model was created using STFT–NMR signals, and product peak intensities were determined by solution NMR. As input data to analyze in GTMR, pseudodata were generated using GMM [50]. In the case of GTMR in the data of cellulose degradation process, the peak of propionate as input data was used, and the peaks of CO_2_ and acetate were predicted as the concentration of production. For plastics analysis, a regression model was created using the STFT–NMR signals and thermal properties. In the case of inverse GTMR, the desired thermal properties were used as input data, and NMR signals were predicted as the higher-order structure descriptors.

## 4. Conclusions

We have developed a solid-state NMR signal deconvolution method using STFT and NTF/NMF, and a prediction method using GTMR. These methods enable 1D solid-state NMR spectra to provide separate signals of multiple phases and components from solid-state NMR spectra. Further, macromolecular samples were characterized, and higher-order structures and thermal properties were predicted. As a new alternative to applying the decoupling to remove anisotropy as unnecessary information in the measurement of ssNMR with a broad line width, signal separation by computational science methods will expand the applicability of low-field ^1^H ssNMR and anisotropic NMR. In the case of NMR data with low digital resolution such as the solid-state NMR and quadrupole nucleus the number of data points can be increased by applying interpolation. In the case of 2D-NMR, it is necessary to use this method by splitting each t1-dimensional FID and creating a series of sub-FIDs. Therefore, these methods will promote data-driven research and development in fields such as machine learning and simulation using ssNMR on macromolecular complexity in materials and foods.

## Figures and Tables

**Figure 1 ijms-22-01086-f001:**
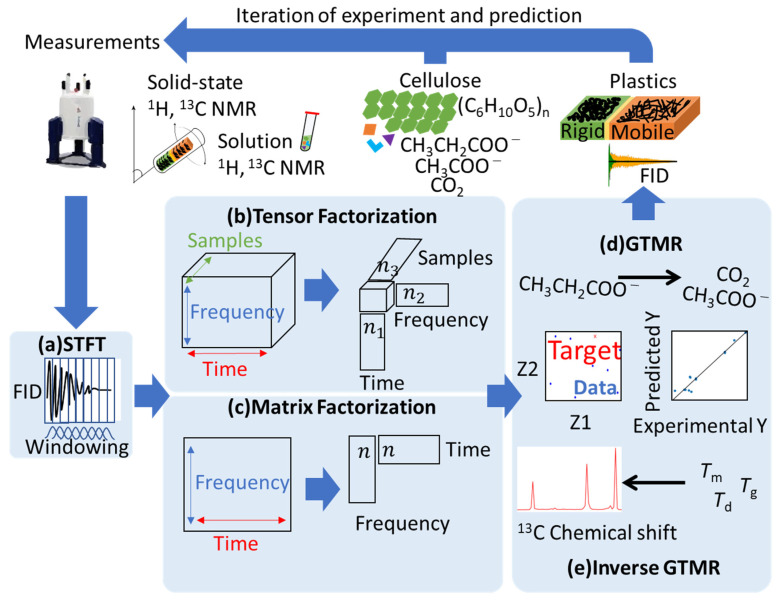
Concept diagram of a material development cycle based on signal deconvolution and prediction for the solid-state nuclear magnetic resonance (ssNMR) of multi-component materials. (**a**) Free induction decay (FID) is transformed into a dataset with time and frequency axes by short-time Fourier transform (STFT). (**b**) In the case of a three-dimensional dataset such as one with multiple samples and conditions, the FID is separated into each component based on the factors of time, frequency, and samples (or condition) by tensor factorization. (**c**) In the case of two-dimensional datasets such as a matrix with time and frequency axes, the FID is separated into each component based on factors of time and frequency by matrix factorization. (**d**) The generative topographic mapping regression (GTMR) accurately predicted the cellulose degradation process shown by catabolic products such as acetate and CO_2_. (**e**) Forward prediction and inverse prediction of GTMR were used to compute the STFT-processed NMR (STFT–NMR) signals from the physical properties of the plastics. This approach is an iterative procedure to achieve convergence between experimental and predicted spectra.

**Figure 2 ijms-22-01086-f002:**
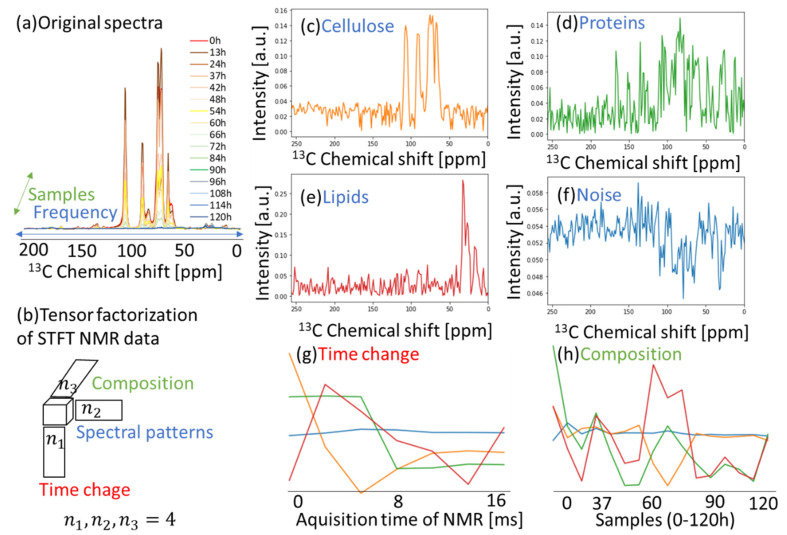
Application of non-negative Tucker decomposition (NTD) to ^13^C cross-polarization–magic-angle spinning (CP-MAS) in the cellulose degradation process. (**a**) Original spectra of ^13^C CP-MAS in cellulose degradation process. (**b**) Tensor factorization of STFT–NMR signals. (**c**–**f**) Spectral patterns (cellulose, lipids, proteins, and noise) when signals were separated into four components. (**g**) Time change of separated components. (**h**) Composition of separated components.

**Figure 3 ijms-22-01086-f003:**
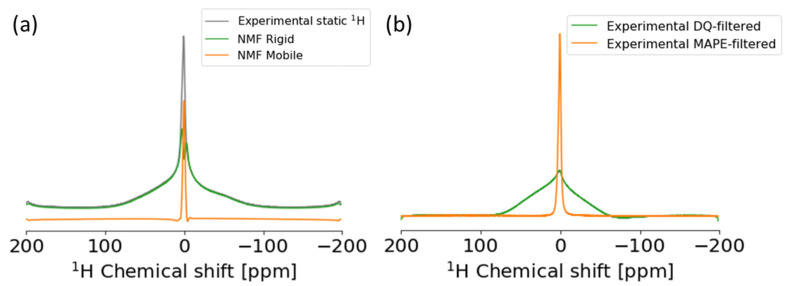
Application of non-negative matrix factorization (NMF) to static ^1^H solid-state NMR of poly-ε-caprolactone (PCL). (**a**) Experimental anisotropic spectrum (gray) and spectra of rigid (green) and mobile (orange) components separated by NMF. (**b**) Experimental spectra of double-quantum (DQ) filtered ssNMR (green) and magic-and-polarization echo (MAPE) filtered ssNMR (orange).

**Figure 4 ijms-22-01086-f004:**
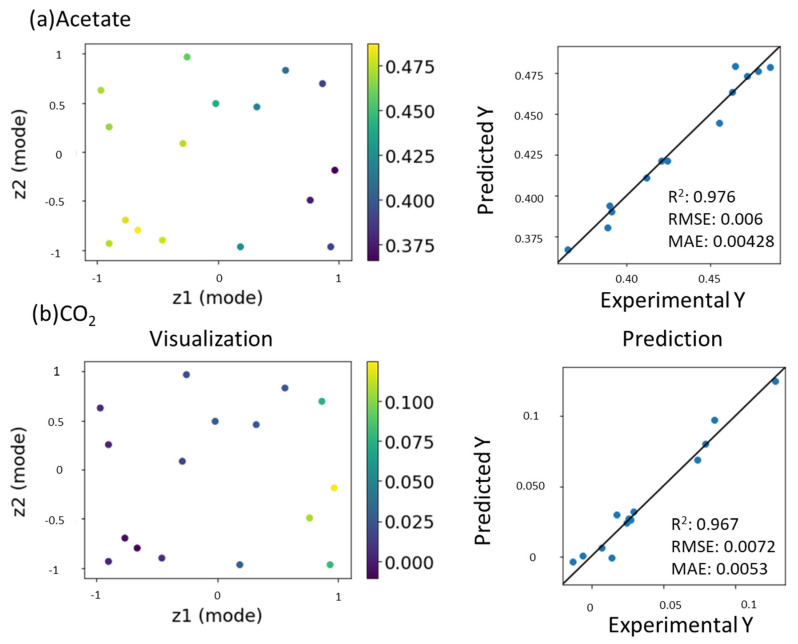
Application of GTMR to NMR data in the cellulose degradation process. (**a**) Visualization and prediction of the concentration of acetate. (**b**) Visualization and prediction of the concentration of CO_2_.

**Figure 5 ijms-22-01086-f005:**
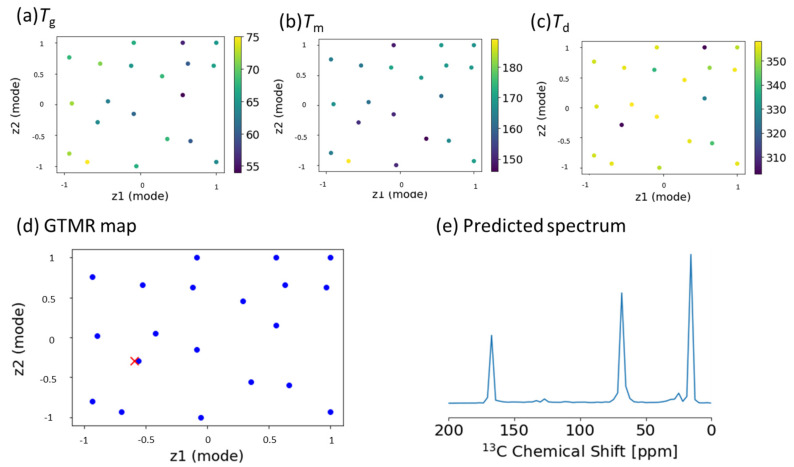
Application of GTMR for predicting NMR data from thermal properties in PLA. (**a**–**c**) *T*_g_, *T*_m_, and *T*_d_ in data map. (**d**) Coordinates corresponding to the target thermal properties in data map. (**e**) Predicted ^13^C CP-MAS spectrum using GTMR.

## Data Availability

The data presented in this study are available on request from the corresponding author.

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
