# Peer review of "Signal Deconvolution and Generative Topographic Mapping Regression for Solid-State NMR of Multi-Component Materials"

_ijms, 2021, doi:10.3390/ijms22031086_

Round 1
Reviewer 1 Report
It is a thorough and detailed work on the development and implementation of techniques for improvement of resolution and to some extent the sensitivity of the solid-state NMR for evaluation of complex solid materials. The authors suggest several new approaches and demonstrate their applicability and advantages on a number of well selected representative models. The manuscript is written in clear and concise language and will be of great help both to NMR spectroscopists specializing in studying complex solid materials, and material chemists, interested in the characterization of novel materials. I find the content and the subject of the manuscript suit well to the International Journal of Molecular Sciences and believe that it can be published 'as is' after minor editorial corrections.
Author Response
[Comments and Suggestions for Authors]
It is a thorough and detailed work on the development and implementation of techniques for improvement of resolution and to some extent the sensitivity of the solid-state NMR for evaluation of complex solid materials. The authors suggest several new approaches and demonstrate their applicability and advantages on a number of well selected representative models. The manuscript is written in clear and concise language and will be of great help both to NMR spectroscopists specializing in studying complex solid materials, and material chemists, interested in the characterization of novel materials. I find the content and the subject of the manuscript suit well to the International Journal of Molecular Sciences and believe that it can be published 'as is' after minor editorial corrections.
[Response]
Thanks for your understanding about techniques for improvement of resolution of the solid-state NMR for evaluation of complex solid materials. We considered responses to all comments, and the revisions of the manuscript are indicated by yellow markers.
Reviewer 2 Report
Dr. Kikuchi, Yamada and Chikayama has presented a methodology paper of ssNMR signal deconvolution and predication. The method has been demonstrated on cellulose, plastics, and algal samples.
This is a major advance as it allows us to analyze the one-dimensional spectra, which are typically limited by resolution, with greater details. We are now able to separate the signals of different components with diverse chemical motifs or dynamics.
The method is of significant interest to the scientific community of NMR spectroscopy as well as polymer and biomolecular research.
The manuscript is well written and logically organized. In addition, the technical details are well presented in the supplementary information file. I believe many scientists (including myself) would benefit substantially from this method.
I only have two minor critiques regarding Fig. 2
1) The noise region in panel 2f is less convincing. The negative peaks indicate that too much signals have been included in the individual biomolecular regions (e.g. cellulose, protein, lipid); therefore, the remaining noise region has negative peaks. Could the authors verify if a better performance could be achieved?
2) Panel 2g and 2h contains important information about the time-dependence of individual components. There are two ambigious aspects about this figure. First what is the x- and y-axis (and quantity and units)? Second, is there any way to provide an estimate of the error margin of each data point in these two plots (e.g. propagated from signal-to-noise ratios)? I can clearly see the significant variation and dependence but I am uncertain about the error bars of the data plotted here.
Author Response
[Comments and Suggestions for Authors]
Dr. Kikuchi, Yamada and Chikayama has presented a methodology paper of ssNMR signal deconvolution and predication. The method has been demonstrated on cellulose, plastics, and algal samples. This is a major advance as it allows us to analyze the one-dimensional spectra, which are typically limited by resolution, with greater details. We are now able to separate the signals of different components with diverse chemical motifs or dynamics. The method is of significant interest to the scientific community of NMR spectroscopy as well as polymer and biomolecular research. The manuscript is well written and logically organized. In addition, the technical details are well presented in the supplementary information file. I believe many scientists (including myself) would benefit substantially from this method.
[Response]
Thanks for your understanding about techniques for ssNMR signal deconvolution and predication. We considered responses to all comments as follows, and the revisions of the manuscript are indicated by yellow markers.
I only have two minor critiques regarding Fig. 2
1) The noise region in panel 2f is less convincing. The negative peaks indicate that too much signals have been included in the individual biomolecular regions (e.g. cellulose, protein, lipid); therefore, the remaining noise region has negative peaks. Could the authors verify if a better performance could be achieved?
[Response]
As a result of verification for better signal separation, of the four separated signals, signals such as cellulose, proteins, and lipids were clearly high-intensity signals, but the noise was relatively low. In order to show the difference in signal and noise intensity, In order to show the difference in signal and noise intensity, the signal is shown in a view of the y-axis (Fig. 2c-f). The revision in line 163 is described as "As a result, of the four signals separated, the cellulose, proteins, and lipids-like signals were clearly high-intensity signals, while the noise was relatively low." and is indicated by a yellow marker.
Figure 2. Application of NTD to 13C CP-MAS in the cellulose degradation process. (a) Original spectra of 13C CP-MAS in cellulose degradation process. (b) Tensor factorization of STFT-NNR signals. (c–f) Spectral features (cellulose, lipids, proteins, and noise) when signals were separated into four components (g) Time change of each component. (h) Composition in each sample.
2) Panel 2g and 2h contains important information about the time-dependence of individual components. There are two ambigious aspects about this figure. First what is the x- and y-axis (and quantity and units)? Second, is there any way to provide an estimate of the error margin of each data point in these two plots (e.g. propagated from signal-to-noise ratios)? I can clearly see the significant variation and dependence but I am uncertain about the error bars of the data plotted here.
[Response]
Panels 2g and 2h have been revised to show details of the x-axis. Since this experiment has not been performed multiple times, it is not possible to add error bars. The signal-to-noise ratio due to factors during NMR measurements is independent of the margin of error of the predicted data points. Computational error information is provided to estimate the margin of error for each data point on these two plots. The revision in line 165 is described as "In the calculation scheme of NTD, convergence tolerance of calculation error is less than 0.001." and is indicated by a yellow marker.

Reviewer 3 Report
In this work, Yamada et al. demonstrate deconvolution and predictive methods to analyze poorly resolved 1D solid-state NMR spectra commonly encountered in the materials sciences. Interpreting these spectra is usually a labor-intensive process and highly susceptible to human bias, errors, and assumptions. Advanced model-free techniques commonly encountered in the computer/data sciences have been gaining popularity in this field for these reasons.
Specifically, Yamada et al. combine short-time Fourier transform (STFT), which effectively separates components based on dynamics (T2*), and non-negative tensor/matrix factorization (NTF/NMF) to deconvolute spectra into distinct components. This is useful especially for reliably quantifying the chemical/morphological species in solid samples. An example of generative topographic mapping regression (GTMR) is also shown to predict STFT, although the benefits/use of this to me was unclear.
Overall, the experimental data has been previously published and the analyses appear to be sound. However, the justification for methods and the information that can be derived from them is poorly discussed. Extensive editing is required for this work to be useful to experimentalists, of whom would be the primary readership.
The following includes a list of points that need to be addressed:
L106 – “For analysis of the 104 three-dimensional dataset of multiple samples and parameters, tensor methods such as canonical polyadic (CP) decomposition[45,46] and Tucker decomposition [47] can be used.” – It is unclear why this is mentioned. Are NTF/NMF superior to these methods? If so, explain why.
L109 – “Prediction of the NMR data and sample properties was calculated using GTMR (Figure 1d,e,S2)” Why must the spectra be predicted? I can understand why the deconvolution by STFT and NTF/NMF is done, but not the GTMR prediction. Is this an iterative procedure to achieve convergence between experimental and predicted spectra? If so, this should be better represented Figure 1 (i.e., showing a cycle).
L112 – “As input data to analyze in GTMR, pseudo data were generated using Gaussian mixture models (GMM) (Figure S3)” – Please explain what is meant by pseudo data.
L115 – “estimate the properties of macromolecules with the desired higher-order structure or physical properties” – Is this just referring to cellulose decomposition as stated in L110? Please provide some examples of what could be meant by “higher-order structure” and “physical properties”
L130 – “Microbial products” does not seem correct. Do the authors mean “catabolic products”?
L139 – (s5a) should be “Figure S5a”
L144 – By what criteria is NCPD not as good as NTD? Please also explain why NCPD and NTD were trialed in this work when the authors appear to favor the NTF/NMF methods. If NCPD and NTD were simply done to compare to NTF/NMF, then this should be discussed in detail at least once in the text.
Figure 2A – A description of the samples are required. Are these timepoints of cellulose decomposition?
Figure 2g and h – Real values are needed for the x and y-axes. What is meant by “Samples” in 2H? This figure needs to be described in much more detail.
Section 2.3 – What does NMF provide beyond just using STFT? Why is NMF used rather than NTF? In fact, I cannot find any demonstration of the NTF method throughout the text.
L176 to L178 – This sentence is hard to understand. Why do the authors consider the Fourier interpolation method to be better than the others?
L184 – What is meant by “regression of NMR data”? As mentioned above, why is the prediction of NMR spectra important?
L191 – “Cellulose is decomposed into propionic acid, acetic acid, and butyric acid.” – In section 2.2, it was stated that cellulose degrades into proteins and lipids. Please clarify this discrepancy.
L198 to L200 – I don’t understand what is meant by this sentence.
Section 2.4 – Overall, the authors need to describe what is being gained by the GTMR. For the degradation of cellulose, what information did GTMR provide? Why is this information important?
Section 3.3 – The use of Python libraries for these analyses it quite useful. The author should comment on the availability of their scripts for people who wish to try out the method.
Author Response
[Comments and Suggestions for Authors]
In this work, Yamada et al. demonstrate deconvolution and predictive methods to analyze poorly resolved 1D solid-state NMR spectra commonly encountered in the materials sciences. Interpreting these spectra is usually a labor-intensive process and highly susceptible to human bias, errors, and assumptions. Advanced model-free techniques commonly encountered in the computer/data sciences have been gaining popularity in this field for these reasons. Specifically, Yamada et al. combine short-time Fourier transform (STFT), which effectively separates components based on dynamics (T2*), and non-negative tensor/matrix factorization (NTF/NMF) to deconvolute spectra into distinct components. This is useful especially for reliably quantifying the chemical/morphological species in solid samples. An example of generative topographic mapping regression (GTMR) is also shown to predict STFT, although the benefits/use of this to me was unclear. Overall, the experimental data has been previously published and the analyses appear to be sound. However, the justification for methods and the information that can be derived from them is poorly discussed. Extensive editing is required for this work to be useful to experimentalists, of whom would be the primary readership.
[Response]
Thanks for your understanding about techniques for deconvolution and predictive methods to analyze poorly resolved 1D solid-state NMR spectra commonly encountered in the materials sciences. We considered responses to all comments as follows, and the revisions of the manuscript are indicated by yellow markers.
The following includes a list of points that need to be addressed:
L106 – “For analysis of the 104 three-dimensional dataset of multiple samples and parameters, tensor methods such as canonical polyadic (CP) decomposition[45,46] and Tucker decomposition [47] can be used.” – It is unclear why this is mentioned. Are NTF/NMF superior to these methods? If so, explain why.
[Response]
We apologize for the incomprehensible description. The generic term including non-negative canonical polyadic decomposition (NCPD) and non-negative Tucker decomposition (NTD) is Non-negative tensor factorization (NTF). The method in Figure 2 is NTD. The method in Figure S6 is NCPD. The revision in line 101 is described as " As algorithms of factorization, in addition to the traditional NMF for analysis of the two-dimensional (2D) dataset, we investigated the application of NTF (non-negative Tucker decomposition (NTD) [45] and non-negative canonical polyadic decomposition (NCPD) [46,47]), which is a factorization algorithm useful for analysis of the 3D dataset of multiple samples and parameters." and is indicated by a yellow marker.
L109 – “Prediction of the NMR data and sample properties was calculated using GTMR (Figure 1d,e,S2)” Why must the spectra be predicted? I can understand why the deconvolution by STFT and NTF/NMF is done, but not the GTMR prediction. Is this an iterative procedure to achieve convergence between experimental and predicted spectra? If so, this should be better represented Figure 1 (i.e., showing a cycle).
[Response]
We apologize for the incomprehensible description. So far, we have been developing methods for routine material evaluation, which produce a lot of NMR spectrum information using NMR including low magnetic field NMR. In this concept, it is necessary to predict the NMR signal using the accumulated NMR data and characteristic information in order to efficiently find the structure with the desired properties. In the actual material development process, this is an iterative procedure to achieve convergence between the experimental and predicted spectra. We revised Figure 1 as a cycle using this method. The revision in line 108 is described as "Furthermore, the high-order structure of materials exerts a significant influence on their macroscopic properties[27]. Traditional design approaches for materials are the experimentally-driven and trial-and-error, are facing significant challenges due to the vast design space of materials. And, computational technologies such as density func-tional theory (DFT)[48] and molecular dynamics (MD)[7], are usually computationally expensive, are difficult to calculate molecular structures from material properties. To address these problems, machine learning assisted materials design is emerging as a promising tool for successful breakthroughs in many areas of science[49]. And, in general, NMR measurement, especially a low magnetic field NMR, is a method for routine material evaluation, produce a lot of NMR dataset[32]. Against this back-ground, in the cycle of developing materials using NMR and other measurements evaluating properties, prediction of the NMR signal using accumulated data of NMR signals and properties is necessary to find a structure with desired properties. In this study, prediction of the NMR data and sample properties was calculated using GTMR (Figures 1d,e,S2) [44]." and is indicated by a yellow marker.
Figure 1. Concept diagram of a material development cycle based on signal deconvolution and prediction for the solid-state NMR of multi-component materials. (a) FID is transformed into a dataset with time and frequency axes by STFT. (b) In the case of a three-dimensional dataset such as one with multiple samples and conditions, the FID is separated into each component based on the factors of time, frequency, and samples (or condition) by tensor factorization. (c) In the case of two-dimensional datasets such as a matrix with time and frequency axes, the FID is separated into each component based on factors of time and frequency by matrix factorization. (d) The GTMR accurately predicted cellulose degradation process shown by catabolic products such as acetic acid and CO2. (e) Forward prediction and inverse prediction of GTMR were used to compute the STFT-processed NMR signals from the physical properties of the plastics. This approach is an iterative procedure to achieve convergence between experimental and predicted spectra.
L112 – “As input data to analyze in GTMR, pseudo data were generated using Gaussian mixture models (GMM) (Figure S3)” – Please explain what is meant by pseudo data.
[Response]
Pseudo data is a dataset with the same distribution as the original dataset generated using GMM. Using GMM, randomly generating data based on the means and covariances will generate new pseudo-data. By performing GTMR calculation based on this, spectrum prediction can be performed as an output. Based on this step, data can be generated without experimentation. The revision in line 123 is described as "And, we also examined the use of pseudo-data as a method of predicting data without going through experiments. Pseudo-data is a dataset with the same distribution as the original dataset generated using Gaussian mixture models (GMM) (Figure S3)[50]. Randomly generating data based on means and covariances using GMM produces new pseudo-data. By performing GTMR calculation from this pseudo-data as input data, spectrum as output can be predicted without preparing new materials." and is indicated by a yellow marker.
L115 – “estimate the properties of macromolecules with the desired higher-order structure or physical properties” – Is this just referring to cellulose decomposition as stated in L110? Please provide some examples of what could be meant by “higher-order structure” and “physical properties”
[Response]
Here, cellulose decomposition is an example of specific data of a polymer material. This method can be applied to other sample systems for pursuing structure-property correlation. Examples of cellulose and plastic will be described as the meanings of "higher-order structure" and "property". In cellulose degradation, "higher-order structure" means the crystalline and amorphous structure of cellulose, and "property" means the product amount of catabolic products. Also, with plastics such as PCL, it is difficult to put into practical use materials with high degradability and poor toughness. In this example, Multiple domain structures with different degrees of entanglement of molecular chains are referred to as "higher-order structures" and thermal and mechanical properties are referred to as "property". In this study, we demonstrate the application of cellulose decomposition and plastic for evaluating our method. The revision in line 130 is described as "This method can be applied to various sample systems for pursuing structure-property correlation. In this study, we demonstrate the application of cellulose degradation and plastic for evaluating our method. Here, in cellulose degradation, "higher-order structure" means the crystalline and amorphous structure of cellulose, and "property" means the product amount of catabolic products. Also, with plastics such as PCL, it is difficult to put into practical use materials with high degradability and poor toughness. In this example, multiple domain structures with different degrees of entanglement of molecular chains are referred to as "higher-order structures" and thermal and mechanical properties are referred to as "property"." and is indicated by a yellow marker.
L130 – “Microbial products” does not seem correct. Do the authors mean “catabolic products”?
[Response]
We revised “Microbial products” to “catabolic products”. The revision in line 154 is indicated by a yellow marker.
L139 – (s5a) should be “Figure S5a”
[Response]
We revised “(s5a)” to “(Figure S5a)”. The revision in line 171 is indicated by a yellow marker.
L144 – By what criteria is NCPD not as good as NTD? Please also explain why NCPD and NTD were trialed in this work when the authors appear to favor the NTF/NMF methods. If NCPD and NTD were simply done to compare to NTF/NMF, then this should be discussed in detail at least once in the text.
[Response]
We apologize for the incomprehensible description. Similar to the response to L106, The generic term including non-negative canonical polyadic decomposition (NCPD) and non-negative Tucker decomposition (NTD) is Non-negative tensor factorization (NTF). The method in Figure 2 is NTD. The method in Figure S6 is NCPD. The criteria that NCPD not as good as NTD is unclear spectral patterns for assigning compounds. The reason that NCPD and NTD were trialed is that NCPD is different from the algorism of NTD using in this work. NTD decomposes the tensor into a small core tensor and factor matrices. NCPD decomposes the tensor into factor matrices without a core tensor. The revision in line 175 is described as " NCPD is different from the algorithm of NTD using in this work. NTD separates the tensor into a small core tensor and factor matrices. NCPD separates the tensor into factor matrices without a core tensor. As a result of separating the spectrum into four components using NCPD, it was not as good as NTD because of the unclear spectral patterns for assigning compounds (Figure S6)." and is indicated by a yellow marker.
Figure 2A – A description of the samples are required. Are these timepoints of cellulose decomposition?
[Response]
The 13C solid-state NMR spectra of the cellulose decomposition process at 16 time points from 0 to 120 hour (gradient representation) are shown in Figure 2a. We described a description of the samples as "This dataset using in figure 2 is a time-dependent dataset of 13C solid-state CP-MAS signals of the cellulose degradation process and also contains signals of metabolic products (proteins and lipids). The 13C ssNMR spectra detect macromolecules of cellulose, proteins and lipids. This dataset is data with frequency and intensity in 16 time points from 0 to 120 hours (Figures 2a)." in line 152.
Figure 2. Application of NTD to 13C CP-MAS in the cellulose degradation process. (a) Original spectra of 13C CP-MAS in cellulose degradation process. (b) Tensor factorization of STFT-NNR signals. (c–f) Spectral features (cellulose, lipids, proteins, and noise) when signals were separated into four components. (g) Time change of each component. (h) Composition in each sample.
Figure 2g and h – Real values are needed for the x and y-axes. What is meant by “Samples” in 2H? This figure needs to be described in much more detail.
[Response]
Panels 2g and 2h have been revised to show details of the x-axis. “Time” in figure 2g means the time of change of the separated signal components. And, “samples” in figure 2h means the samples at each measurement time from 0 to 120 hours of 13C CP-MAS NMR. Unit of y-axis is not needed since it is arbitrary (arbitrary unit, a.u.). The revision in line 161 is described as "Time in figure 2g means the time of change of the separated signal components. And, samples in figure 2h means the sample at each measurement time from 0 to 120 hours of 13C CP-MAS NMR." and is indicated by a yellow marker.
Section 2.3 – What does NMF provide beyond just using STFT? Why is NMF used rather than NTF? In fact, I cannot find any demonstration of the NTF method throughout the text.
[Response]
NMF provides hidden patterns from axes of time and frequency about each frequency in each time segment created by STFT, is able to separate NMR signals to multiple-component with different T2*. NMF is a method for a dataset of two-dimensions, NTF is a method for a dataset of three-dimensions. In section 2.3, we demonstrated the application of NMF to single data of PCL. The revision in line 194 is described as "In this study, we demonstrated the application of NMF to 2D dataset created from single data of PCL using STFT. Unlike NTF for 3D dataset mentioned above, NMF is a method for 2D dataset. NMF provides hidden patterns from axes of time and frequency about each frequency in each time segment created by STFT, is able to separate NMR signals to multi-component with different T2*. It was shown that by using NMF, rigid and mobile phases can be extracted from a broad static 1H anisotropic spectrum of PCL as components related to different physical properties(Figures 1c,3)." and is indicated by a yellow marker.
L176 to L178 – This sentence is hard to understand. Why do the authors consider the Fourier interpolation method to be better than the others?
[Response]
When using this method for NMR data with low digital resolution such as solid-state NMR and quadrupole nucleus, this signal deconvolution method needs additional efforts. We demonstrated some interpolation methods for increasing data points. The Fourier interpolation method provides an interpolated spectrum without artifact signals. the spectrum interpolated by other methods has an artifact in the extended region. The revision in line 231 is described as " When using this method for NMR data with low digital resolution such as solid-state NMR and quadrupole nucleus, this signal deconvolution method needs additional efforts. We demonstrated some interpolation methods for increasing data points (Figure S8). The Fourier interpolation method provides an interpolated spectrum without artifact signals. Spectra interpolated by other methods have artifacts in the extended region." and is indicated by a yellow marker.
L184 – What is meant by “regression of NMR data”? As mentioned above, why is the prediction of NMR spectra important?
[Response]
“regression of NMR data” mean prediction of NMR spectra. NMR signals are rich in information about the structural features of the molecules contributing to their physical/chemical/biological properties, has potential for use as a descriptor in quantitative structure–activity/property relationship modeling studies. Therefore, the prediction of NMR signals is important for developing materials. The revision in line 225 is described as " Chemical shifts of NMR are rich in chemical information and enable encoding the structural features of the molecules contributing to their physical/chemical/biological properties. Thus, it has potential for use as a descriptor in quantitative structure–activity/property relationship (QSAR/QSPR) modeling studies[49]. GTMR was applied for analyzing these studies. Therefore, prediction of NMR signals is important for developing materials. This study is the first application of GTMR for prediction of NMR signals." and is indicated by a yellow marker.
L191 – “Cellulose is decomposed into propionic acid, acetic acid, and butyric acid.” – In section 2.2, it was stated that cellulose degrades into proteins and lipids. Please clarify this discrepancy.
[Response]
In figure 2, 13C ssNMR spectra detect macromolecules of cellulose, proteins and lipids. On the other hand, liquid NMR of cellulose in section 2.4, detect small molecules such as propionic acid, acetic acid, and butyric acid. In cellulose decomposition, cellulose is metabolized for microbial cell components such as proteins and lipids and produce short-chain fatty acids. The revision in line 231 is described as "In the degradation of cellulose, cellulose is metabolized into microbial cell components such as proteins and lipids and catabolized into short-chain fatty acids. In figure 2, macromolecules (cellulose, proteins and lipids) were detected using the solid 13C spectrum. In addition, to track the process of material degradation, using liquid NMR spectra detect small molecules such as propionic acid, acetic acid, and butyric acid." and is indicated by a yellow marker.
L198 to L200 – I don’t understand what is meant by this sentence.
[Response]
When cellulose is finally catabolized to CO2 by the catabolism of microorganisms, it is decomposed into acetic acid with two carbon atoms and CO2 with one carbon atom via propionic acid with three carbon atoms. Using the peak intensities of propionic acid as input data, it was possible to predict the peak intensities of acetic acid and CO2, which are indicators of the amount of production. The revision in line 240 is described as "When cellulose is finally catabolized to CO2 by the catabolism of microorganisms, it is metabolized into acetic acid with two carbon atoms and CO2 with one carbon atom via propionic acid with three carbon atoms. When the peak intensity of propionic acid is used as input data of GTMR, it was possible to predict the acetic acid (Figure 4a right ; R2 = 0.947) with the two carbon in the previous stage of the final product, and CO2 (Figure 4b right; R2 = 0.948) with one carbon in the final product." and is indicated by a yellow marker.
Section 2.4 – Overall, the authors need to describe what is being gained by the GTMR. For the degradation of cellulose, what information did GTMR provide? Why is this information important?
[Response]
For analyzing the degradation of cellulose, GTMR provides information about predicted NMR signals of products in cellulose decomposition. This information is important for monitoring due to a key component in compound production using cellulose. However, as mentioned above, the data analysis of the cellulose decomposition process is one example, and we believe that the GTMR prediction method can also analyze the structural property correlation of various plastic materials. The revision in line 246 is described as "For analyzing the degradation process of cellulose, GTMR provides information about predicted NMR signals of products in cellulose degradation. This information is important for monitoring of this process due to a key component in compound production using cellulose." and is indicated by a yellow marker.
Section 3.3 – The use of Python libraries for these analyses it quite useful. The author should comment on the availability of their scripts for people who wish to try out the method.
[Response]
Information about the use of Python libraries for these analyses is available at “http://dmar.riken.jp/NMRinformatics/”. The revision about this point in the section of Supplementary Materials is described as " Python tools developed in this study are available at http://dmar.riken.jp/NMRinformatics/" and is indicated by a yellow marker.
